# Ginseng-Induced Changes to Blood Vessel Dilation and the Metabolome of Rats

**DOI:** 10.3390/nu12082238

**Published:** 2020-07-27

**Authors:** Hyeon-Jeong Lee, Bo-Min Kim, Soo Hee Lee, Ju-Tae Sohn, Jae Woong Choi, Chang-Won Cho, Hee-Do Hong, Young Kyoung Rhee, Hyun-Jin Kim

**Affiliations:** 1Division of Applied Life Sciences (BK21 plus), Gyeongsang National University, 501 Jinjudae-ro, Jinju, Gyeongsangnam-do 52828, Korea; lhj051592@naver.com (H.-J.L.); bomin95@hanmail.net (B.-M.K.); 2Department of Anesthesiology and Pain Medicine, Gyeongsang National University School of Medicine, Gyeongsang National University Hospital, 79 Gangnam-ro, Jinju, Gyeongsangnam-do 52727, Korea; lishiuji@naver.com (S.H.L.); jtsohn@gnu.ac.kr (J.-T.S.); 3Institute of Health Sciences, Gyeongsang National University, 501 Jinjudae-ro, Jinju, Gyeongsangnam-do 52828, Korea; 4Research Group of Traditional Food, Korea Food Research Institute, Wanju-gun, Jeollabuk-do 55365, Korea; choijw@kfri.re.kr (J.W.C.); cwcho@kfri.re.kr (C.-W.C.); honghd@kfri.re.kr (H.-D.H.); 5Department of Food Science & Technology, and Institute of Agriculture and Life Science, Gyeongsang National University, 501 Jinjudaero, Jinju, Gyeongsangnam-do 52828, Korea

**Keywords:** blood vasodilation, ginseng, lysophosphatidylcholine, metabolomics, steroid hormones

## Abstract

Ginseng consumption has been shown to prevent and reduce many health risks, including cardiovascular disease. However, the ginseng-induced changes in biofluids and tissue metabolomes associated with blood health remain poorly understood. In this study, healthy rats were orally administered ginseng extracts or water for one month. Biofluid and tissue metabolites along with steroid hormones, plasma cytokines, and blood pressure factors were determined to elucidate the relationship between ginseng intake and blood vessel health. Moreover, the effect of ginseng extract on blood vessel tension was measured from the thoracic aorta. Ginseng intake decreased the levels of blood phospholipids, lysophosphatidylcholines and related enzymes, high blood pressure factors, and cytokines, and induced vasodilation. Moreover, ginseng intake decreased the level of renal oxidized glutathione. Overall, our findings suggest that ginseng intake can improve blood vessel health via modulation of vasodilation, oxidation stress, and pro-inflammatory cytokines. Moreover, the decrease in renal oxidized glutathione indicated that ginseng intake is positively related with the reduction in oxidative stress-induced renal dysfunction.

## 1. Introduction

Ginseng (*Panax ginseng* Meyer) has been widely used as a pharmaceutical plant for thousands of years owing to its various well-established health benefits. Accumulating evidence suggests that various ginseng compounds, including ginsenosides, polysaccharides, vitamins, essential oils, and phenolic compounds [1], are positively associated with many of the observed health benefits such as fatigue alleviation, obesity mitigation, diabetes control, immunoregulation, and an atheroprotective effect [2]. In particular, many factors related with cardiovascular functions are affected by ginseng intake and its compounds. Although previous studies demonstrated that ginseng intake ameliorated arterial stiffness [3] and blood pressure [4,5], inhibited the progression of atherosclerotic lesions [6], regulated blood lipid profiles, and inhibited inflammatory factors [7], the factors driving any correlation between ginseng intake and cardiovascular functions are still not completely understood.

Recently, omics technologies, including genomics, proteomics, and metabolomics, have been applied to better understand the pathogenesis of cardiovascular diseases [8,9,10] and to find new biomarkers for diagnosis or treatment. In particular, metabolomics studies demonstrated that branched-chain amino acids, short-chain dicarboxylacylcarnitine species, and trimethylamine N-oxide were strongly associated with cardiovascular risks [8]. Moreover, metabolomics studies on ginseng intake revealed that a decrease in blood lysophosphatidylcholine (LPC) levels in patients with prehypertension [5] and rats [11] might reduce blood pressure. Moreover, epidemiological data have indicated that estrogen hormone levels are strongly related to the progression of cardiovascular diseases [12]; however, the specific ginseng intake-induced changes in metabolite profiles, including any changes to steroid hormones with atheroprotective effects, have not been clearly demonstrated.

Besides its use in the treatment of certain diseases, many healthy people also incorporate ginseng as a dietary supplement to maintain their health; however, most of the available ginseng studies have been conducted using disease animal models and patients with prehypertension, and the influence of ginseng has not been comprehensively assessed in a healthy model. Therefore, in this study, global and targeted metabolite profiles of the biofluids and tissues from healthy rats fed ginseng were analyzed using ultra-high performance liquid chromatography (UPLC)-quadrupole (Q)-time-of-flight (TOF) mass spectrometry (MS), and various molecular markers associated with blood pressure and inflammation were investigated to better understand the relationship between ginseng intake and general blood vessel health.

## 2. Materials and Methods

### 2.1. Ginseng Extract

The ginseng water extract was kindly provided by the Korean Food Research Institute. The composition of the ginsenosides of the extract, determined by HPLC (Shimadzu, Tokyo, Japan), is shown in Appendix A.

### 2.2. Animals and Ginseng Administration

All animal experiments were approved by the committee on the Ethics of Animal Experiments of Gyeongsang National University (Permit Number: GNU-151208-M0067). Six-week-old male Sprague–Dawley rats purchased from Koateck (Pyeongtaek, Korea) were randomly housed in cages in a controlled environment under room temperature (24 ± 1 °C) with a 12-h light-dark cycle. The rats were fed a commercial diet (Envigo 2018S, Koatech) and provided tap water ad libitum. After 2 weeks, the rats were divided into three groups, control, low-dose ginseng extract (GL), and high-dose ginseng extract (GH), with 10 rats per group. All groups were fed the normal diet, and rats in the GL and GH groups were orally administered 100 mg/kg and 200 mg/kg of ginseng extract per day, respectively, for 1 month. The control group was orally administered water (2 mL). Food intake and the body weight of the rats were measured weekly. Plasma was collected from the postcaval vein after sacrifice. Liver, kidney, and epididymal adipose tissues were weighed and immediately frozen in liquid nitrogen. Urine was collected using a metabolic cage with 1 mL of 1% sodium azide (Sigma-Aldrich, St. Louis, MO, USA). All samples were stored at −80 °C until analysis.

### 2.3. Blood Lipid Characteristics

Triglyceride (TG), blood total cholesterol (TC), high density lipoprotein (HDL)-cholesterol, and low density lipoprotein (LDL)-cholesterol levels in the plasma were measured using a clinical chemistry analyzer (Fuji Film Co., Tokyo, Japan). The plasma oxidized LDL (oxLDL) level was analyzed using an ELISA kit (SEA527Ra, Cloud-Clone Corp., Houston, TX, USA).

### 2.4. Global Metabolomics Analysis

The tissue and urine samples were lyophilized. Plasma was precipitated with cold acetone. After centrifugation, the supernatants were dried by a Centrivap Speedvac concentrator (Labconco Co., Kansas City, MO, USA). The residues were resolved by 20% methanol with an internal standard (terfenadine). Lyophilized liver and kidney samples were homogenized with 1 mL of 50% methanol containing terfenadine as an internal standard using a bullet blender (Next Advance, Troy, NY, USA). Freeze-dried urine was dissolved in distilled water with 8-bromoguanosine as an internal standard. After centrifugation, all sample supernatants were analyzed by UPLC-Q-TOF MS (Waters, Milford, MA, USA).

Plasma, liver, kidney, and urine metabolite profiles were analyzed with UPLC-Xevo-Q-TOF-MS equipped with an Acquity UPLC BEH C_18_ column (2.1 × 100 mm, 1.7 μm; Waters) at a column temperature of 40 °C. Mobile phases comprised 0.1% formic acid in water (A) and 0.1% formic acid in acetonitrile (B) at a flow rate of 0.35 mL/min for 12 min. The eluted metabolites were analyzed using Q-TOF MS in positive ESI mode. The capillary and sampling cone voltages were set at 3 kV and 30 V, respectively. The cone gas and desolvation flow rate were 30 L/h and 800 L/h, respectively, and desolvation and source temperatures were set to 400 °C and 120 °C, respectively. The TOF MS data were collected in the range of 50 to 1500 m/z with a scan time of 0.2 s. Leucine–enkephalin ([M + H] = 556.2771 Da) was used as a reference compound for lock mass at a frequency of 10 s to maintain analytical accuracy and reproducibility. A quality control sample composed of the mixture of all samples was analyzed once every 10 samples.

MassLynx software (Waters) was used for the collection, normalization, and alignment of the datasets obtained for UPLC-Q-TOF-MS. Data were collected and aligned in an m/z range of 50–1500, *m* and *z* width of 0.05, retention time width of 0.2, peak width of 5%, peak-to-peak baseline noise of 5000, noise elimination level of 6, and marker intensity threshold of 30,000. The collected and aligned data were normalized by each internal standard. Metabolites were identified by online databases, including Human Metabolome, Metlin, and MassBank, along with reference to the literature and authentic standards.

### 2.5. Steroid Hormone Analysis

For urinary steroid hormone analysis, the lyophilized urine (30 mg) was dissolved in 150 μL of 80% methanol with 17β-estradiol-d5 (CDN Isotopes Inc., pointe-Claire, QC, Canada) as an internal standard. After sonication and centrifugation, the supernatant was used for urinary steroid hormone analysis. For plasma steroid hormone analysis, 200 μL of plasma was mixed with 600 μL of methanol containing the internal standard. After mixture and centrifugation, the supernatants were dried, and the residue was dissolved in 100 μL of methanol prior to injection. Steroid hormone profiles of plasma and urine were analyzed with a Vion UPLC-Q-TOF-MS system (Waters) in positive multiple reaction monitoring (MRM) mode. Precursor and product ions of each steroid hormone were used for analysis (Appendix A). The column and LC/MS analysis conditions were the same as those described above for the global metabolite analysis with minor modifications. In particular, the cone gas and desolvation flow rate were 30 L/h and 800 L/h, respectively. Data processing of the MRM data was conducted with UNIFI software. All MRM data were integrated by retention time and the precursor and product ion m/z values for quantification. All mass data were normalized using the internal standards. Identification of the metabolites was based on MRM data, data from previous studies [13], and authentic standards of steroid MS/MS data.

### 2.6. Analysis of Plasma Cytokines and Enzymes

Activities of plasma LPC in generating the enzymes lipoprotein-associated phospholipase A_2_ (Lp-PLA_2_) and lecithin cholesterol acyltransferase (LCAT) were determined using ELISA kits (Cusabio, Houston, TX, USA) according to the manufacturer’s instructions. Plasma inflammatory cytokines, including IFN-γ, IL-1β, IL-6, and tumor necrosis factor-alpha (TNF-α) were measured using a Luminex screen assay kit (R&D Systems, Minneapolis, MN, USA) and a Luminex analyzer (Luminex, Austin, TX, USA). The inhibitory effects of ginseng on angiotensin-converting enzyme (ACE) and plasma ACE activity were measured with the Cushman and Cheung method [14] with minor modifications. The plasma angiotensin II concentration was determined using ELISA (Sigma-Aldrich, St. Louis, MO, USA).

### 2.7. Blood Vessel Tension Measurement Using an Organ Chamber

The aortic rings were isolated and prepared for tension measurements as previously described [15]. The rats of each group were sacrificed by inhalation of 100% carbon dioxide. The descending thoracic aorta was removed and cut to a length of 2.5 mm. The endothelium of the isolated rat aorta was removed by rolling the aortic rings with two 25-gauge needles forward and backward. The aortic ring was suspended in Grass isometric transducers (FT-03, Grass Instrument, Quincy, MA, USA) under a resting tension of 3.0 *g* in a 10 mL Krebs organ bath maintained at 37 °C. The aorta was continuously aerated using 95% O_2_ and 5% CO_2_ to maintain pH values from 7.35 to 7.45. A 3.0-*g* resting tension was used to equilibrate the rings for 120 min, and the bath solution was changed every 30 min. To confirm the intact endothelium, phenylephrine (0.1 μM) was added to the organ bath with endothelium-intact aortic rings and the production of a sustained and stable contraction of phenylephrine was verified. Next, acetylcholine (10 μM) was added to the organ bath containing aortic rings with phenylephrine-induced contraction, and the endothelium-intact aorta was confirmed by observing more than 85% acetylcholine-induced relaxations from the phenylephrine-induced contractions. To confirm the endothelial denudation, phenylephrine (10 nM) was added to the organ bath with the endothelium-denuded aortic rings, which produced a sustained and stable contraction. Acetylcholine (10 μM) was then added to the organ bath containing the aortic rings with phenylephrine-induced contraction, and endothelial denudation was confirmed by observing less than 15% acetylcholine-induced relaxation from phenylephrine-induced contraction. The isolated aortic rings showing acetylcholine-induced relaxation from phenylephrine-induced contraction were washed with fresh Krebs solution to restore the baseline resting tension.

We next examined the effect of ginseng extract on the isolated rat aorta with or without the endothelium and on the isolated endothelium-intact rat aorta pretreated with the nitric oxide synthase (NOS) inhibitor N^W^-nitro-L-arginine methyl ester (L-NAME, 10^−4^ M). The isolated endothelium-intact rat aorta was pretreated with L-NAME (10^−4^ M) for 20 min. After 10^−6^ and 10^−7^ M phenylephrine produced vasoconstriction in the endothelium-intact aorta with or without L-NAME and in the endothelium-denuded aorta, respectively, ginseng extract (10^−5^ to 10^−1^ mg/mL) was cumulatively added to the organ bath, and dose–response curves induced by ginseng extract were constructed.

### 2.8. Statistical Analysis

Pearson’s correlation coefficients between variables were calculated and visualized by R software. Heatmaps were visualized using R with the gplots package to evaluate relationships among metabolites and blood vessel health-improving factors such as plasma lipid oxidation metabolites, including LPC and its related enzyme activities, HDL, oxLDL, inflammation, and blood pressure. SIMCA-P^+^ version 12.0.1 (Umetrics, Umeå, Sweden) was used to analyze global metabolites and steroid hormones processed by MarkerLynx and UNIFI, respectively. Partial least-squares discriminant analysis (PLS-DA) was used to visualize the differences among sample groups. All data, including metabolites, steroid hormones, blood and basic characteristics, and enzyme activities, were statistically analyzed by one-way analysis of variance (ANOVA) with Duncan’s test (*p* < 0.05) using SPSS 17.0 (SPSS Inc., Chicago, IL, USA). The effect of ginseng extract-induced vasodilation was analyzed using two-way repeated-measures ANOVA followed by Bonferroni’s multiple comparison test.

## 3. Results

### 3.1. Animal Characteristics

The animal characteristics of the three groups in terms of body weight gain, tissue weight, TG, and cholesterol are summarized in Table 1. There was no change in the body weight gain; adipose tissue, kidney, and liver weights; or TG level in all groups (control, GL, and GH). However, TC (*p* = 9.94 × 10^−6^), HDL (*p* = 3.64 × 10^−7^), and oxLDL (*p* = 1.61 × 10^−5^) levels decreased by about 17%, 29%, and 40%, respectively, in the GH group compared to those of the control (Table 1), while the LDL and oxLDL level of the GL and GH groups decreased by about 10% and 40% compared to those of the control. In addition, the oxLDL/HDL (*p* = 5.78 × 10^−4^) ratio of the GL group was 42% lower than that of the control, while the oxLDL/LDL (*p* = 7.93 × 10^−4^) and oxLDL/TC (*p* = 7.33 × 10^−5^) ratios of both the GL and GH groups were about 30% lower than those of the control. The ratio of total to HDL cholesterol was not affected by the treatments.

### 3.2. Metabolomic Analysis

The differences in UPLC-Q-TOF MS metabolite profiles of the plasma, urine, kidney, and liver from the control, GL, and GH groups were visualized and statistically analyzed with PLS-DA (Figure 1, Appendix A). There was no significant clustering of urine and liver metabolites, nor clear separation among groups (Appendix A); however, untargeted plasma, kidney, and targeted plasma and urine steroid hormones were clearly separated along the first two-component PLS-DA score plots with statistically acceptable quality parameters (R_2_X > 0.43, R_2_Y > 0.57, and Q_2_ > 0.47; *p*-value < 0.0001). The cross-validation values (R_2_ intercept < 0.10 and Q_2_ intercept < −0.29) analyzed by the permutation test (*n* = 200) indicated that the PLS-DA models for plasma, kidney, and steroid hormones were statistically acceptable. To identify the metabolites contributing to these observed differences, the *p*-values of all normalized chromatogram intensities of plasma and kidney metabolites and steroid hormones were analyzed. Moreover, urine and liver metabolites were statistically analyzed to find the metabolites affected by the ginseng intake, although there was no significant difference among sample groups on the PLS-DA score plots.

Nine plasma metabolites [LPCs (16:0, 16:1, 17:0, 18:0, 18:1, 18:2, 18:3, and 20:4) and lysophosphatidylethanolamine (LPE) (20:4)], six kidney metabolites (uracil, oxidized glutathione, succinyladenosine, riboflavin, leucyl histidine, and stearoylcarnitine) and three liver metabolites [carnitine, LPE (20:4), and LPC (18:1)] were identified to have the greatest contributions to the effect of ginseng. In addition, four plasma steroid hormones (estradiol-3-glucuronate, estrone-3-hemisuccinate, 11α-hydroxyestradiol, and 21α-hydroxyprogesterone) and eight urinary steroid hormones (27-hydroxycholesterol, metandienone, 2-hydroxyestrone-1+4-N-acetylcysteine, 6-dehydroestradiol diacetate, estriol-16,17-diacetate, 2,3-dimethoxyestradiol, 5-androstenediol, and 11β-hydroxyestradiol derivatives) were identified as the greatest contributors to the observed differences among groups (Table 2).

The normalized chromatogram intensities of all identified metabolites and steroid hormones were relatively compared (Appendix A) and their fold changes were calculated (Table 2). Interestingly, in plasma, ginseng intake significantly decreased the levels of LPCs and LPE (18:3) in a concentration-dependent manner, and their levels in the GH group were about 2–7 times lower than those of the control. In the liver, the levels of acetylcarnitine and LPE (20:4) were slightly increased, while the LPC (18:1) levels were similar between the control and ginseng intake groups. In the kidney, ginseng intake decreased the relative abundance of oxidized glutathione, uracil, and leucyl histidine, whereas stearoylcarnitine and riboflavin were increased by GH. In particular, the oxidized glutathione level of the GL and GH groups was about two times lower than that of the control.

The majority of the identified plasma and urinary steroid hormones were altered by GL compared to those of the control, but their levels were mostly recovered by GH except for hydroxyestrone-1+4-N-acetylcysteine, 6-dehydroestradiol diacetate, 5-androstenediol, and 11β-hydroxyestradiol derivatives. Among the urine steroid hormones, the hydroxyestrone-1+4-N-acetylcysteine level increased about two times by GL and GH compared to those of the control, whereas the 11β-hydroxyestradiol derivatives level increased about 6.6 times by GL and GH. By contrast, the 5-androstenediol level decreased by about 30% in the GL and GH groups.

### 3.3. Blood Vessel Health-Related Factors

The effects of ginseng intake on blood LPC-generated enzymes (Lp-PLA_2_ and LCAT), blood cytokines (IFN-γ, IL-1β, IL-6, and TNF-α), and blood pressure factors (ACE activity and angiotensin II of blood and ACE inhibitory activity) are summarized in Figure 2. The Lp-PLA_2_ activity of the GH group was reduced by 25% compared to that of the control, while the LCAT activity of the GL and GH groups was significantly reduced by about 7% and 9%, respectively. In addition, the levels of blood cytokines were decreased by ginseng intake, although the decrease in the IFN-γ level was not statistically significant. The levels of IL-1β, IL-6, and TNF-α of the GH group were decreased by 54%, 13%, and 31%, respectively, compared to those of the control. The in vitro ACE inhibitory activity of the ginseng extract treatment increased in a concentration-dependent manner, and 10 mg of the extract showed 85% inhibitory activity. The plasma ACE activity and angiotensin II concentrations of the GL and GH groups were reduced by about 23% and 22%, respectively, compared to those of the control.

### 3.4. Correlation Analysis

The correlations among all identified metabolites, steroid hormones, and blood vessel health-related factors, including TC, HDL, oxLDL, angiotensin II, IL-1β, IL-6, TNF-α, Lp-PLA_2_, and LCAT, were visualized with a correlation matrix (Figure 3). Most of the identified LPCs, including LPE (18:3), uracil, oxidized glutathione, leucyl histidine, and 5-androstenediol, were positively correlated with most of the blood vessel health-related factors. In particular, blood LPCs showed strong positive correlations with these factors (0.45 ≤ r^2^ ≤ 0.8), while liver LPC (18:1), estradiol-3-glucuronate, estrone-3-hemisuccinate, 21 α -hydroxyprogesterone, 2,3-dimethoxyestradiol, estriol-16,17-diacetate, and metandienone were only positively correlated with HDL, Lp-PLA_2_, TNF-α, and TC. However, riboflavin, acetylcarnitine, steoylcarnitine, liver LPE (20:4), 27-hydroxycholesterol, 6-dehydroestradiol diacetate, 2-hydroxyestrone-1+4-N-acetylcysteine, 11β-hydroxyestradiol derivatives, and 11α-hydroxyestradiol were negatively correlated with all of the analyzed factors. In particular, 11β-hydroxyestradiol derivatives had a strong negative correlation with ox-LDL (r^2^ = −0.94), IL-1β (r^2^ = −0.85), and IL-6 (r^2^ = −0.84) (Appendix A).

### 3.5. Effect of Ginseng Extract on Blood Vessel Tension

The effect of ginseng extract on blood vessel tension was measured from the thoracic aorta (Figure 4). As shown in Figure 4A,Ba, the ginseng extract (10^−2^ and 10^−1^ mg/mL) produced vasodilation in the endothelium-intact aorta by 13% and 40%. However, the highest concentration of ginseng extract (10^−1^ mg/mL) only produced vasodilation in the endothelium-denuded aorta and in the endothelium-intact aorta pretreated with the NOS inhibitor L-NAME by 10% (Figure 4Bb) and by 9% (Figure 4Bc), respectively.

## 4. Discussion

In this study, the effects of ginseng intake on the physiological characteristics and metabolite profiles, including steroid hormones, from healthy male rats were investigated. Moreover, chemical, physiological, and enzymatic parameters associated with vasodilation of the blood vessel were investigated. Some of the metabolites and parameters found to be influenced by ginseng have been reported in previous studies, while others were newly found in this study.

Overall, ginseng intake (100 mg/kg or 200 mg/kg of body weight) in rats reduced blood lipid parameters, including TC, HDL, LDL, and oxLDL, resulting in a decrease in the ratios of oxLDL/HDL, oxLDL/LDL, and oxLDL/TC, which are typical lipid biomarkers used for evaluating oxidation and anti-oxidation status in type 2 diabetes mellitus [16]. However, the body weight and the ratio of total to HDL cholesterol, a specific marker of coronary artery disease [17], were not influenced by ginseng intake. Moreover, interestingly, these blood lipid parameters were strongly correlated with the levels of blood phospholipids such as LPCs and LPE (C18:3), and with renal oxidized glutathione, and ginseng intake reduced these levels in a concentration-dependent manner. This result is in agreement with a recent report showing that blood LPC levels of prehypertensive patients that consumed ginseng were lower than those of a placebo group [5]. Similarly, daily intake of 300 mg/kg of ginseng was previously reported to decrease plasma LPC levels in Sprague–Dawley rats [11].

LPC, a major phospholipid of oxLDL that is closely related with the tissue damage caused by atherosclerosis [18,19] is generally produced under various physiological conditions by endogenous LP-PLA_2_ and LCAT-mediated hydrolysis of lipoproteins [19]. LPCs play numerous important physiological and pathophysiological roles, including in vascular development, reproduction, and myelination, along with associations with neuronal diseases and cancer, in both humans and animals. In particular, accumulating evidence indicates that elevated levels of LPCs, present in oxidatively damaged oxLDL, are linked to the cardiovascular diseases associated with diabetes, atherosclerosis, ischemia, and renal failure [20]. Moreover, LPCs activate inflammatory pathways and have an atherogenic effect on oxLDL and vascular smooth muscle cells, interrupt endothelial nitric oxide production, deteriorate the endothelium-dependent relaxations regulated by endothelium-derived relaxing factors, and directly adjust contractile reactions in vascular smooth muscles [21].

To clarify the mechanism contributing to the observed decrease in blood LPCs by ginseng intake, we investigated the blood LPC-generating enzymes’ activities; those of LCAT and Lp-PLA_2_. Ginseng intake reduced the activities of both enzymes, resulting in an overall decrease in the level of blood LPCs. These results can support the positive correlation between ginseng consumption and its efficacy on major cardiovascular risk factors such as hypertension, cardiac disease, and hyperlipidemia [22,23,24].

Along with major changes in blood LPCs, some blood and urinary hormones, mostly estrogen hormones, were also significantly affected by ginseng intake, although their quantitative changes were not highly significant, except for 2-hydroxyestrone-1+4-N-acetylcysteine and 11β-hydroxyestradiol derivatives from urine. Recent studies suggested that ginsenosides, combined with estrogen receptors, regulated the biological effect of estrogen hormones [25]. In particular, estradiol and its metabolite, hydroxyestradiol, highly stimulated endothelial nitric oxide production [12], inhibited the proliferation and collagen synthesis in rat cardiac fibroblasts, and prevented oxLDL formation [26].

These results revealed that the reduction in blood phospholipids and increased estrogen metabolites, including 11β-hydroxyestradiol derivatives, by ginseng intake might be positively associated with blood vessel health through prevention of inflammation and oxidative stress, with concomitant stimulation of nitric oxide-related vasodilation. Indeed, we found that ginseng intake decreased the levels of cytokines, including IL-1β, IL-6, and TNF-α, and factors associated with high blood pressure, including ACE activity and angiotensin II. Consistent with previous reports [27,28], ginseng extract also produced vasodilation via endothelial nitric oxide activation. The maximal vasodilation induced by ginseng extract was higher in the endothelium-intact aorta (40%) than in the endothelium-denuded (10%) or L-NAME-pretreated endothelium-intact aorta (9%), suggesting that the vasodilation induced by ginseng extract is dependent on endothelial nitric oxide (Figure 4).

Moreover, blood LPCs and LPE (18:3) showed strong positive correlations with blood vessel health-related factors, whereas estrone metabolites, including 2-hydroxyestrone-1+4-N-acetylcysteine and 11β-hydroxyestradiol derivatives, had strong negative correlations with these factors. These results suggested that ginseng improved blood vessel health through the protection of oxidative stress, inflammation, and high blood pressure factors. Ginseng-derived compounds and/or ginseng-influenced endogenous metabolites such as LPCs, LPE (18:3), and estrogen metabolites are directly and/or indirectly associated with the improvement of blood vessel health. In particular, the analysis of blood vessel tension clearly indicated that ginseng had a vasodilation effect through activation of endothelial nitric oxide synthase (eNOS) and the inhibition of ACE and angiotensin II activities [29,30], which are related to hypertension; however, the effects of ginseng and individual compounds on eNOS activity were not investigated in the present study.

These results are in agreement with previous studies that suggested a close association of LPCs and 11β-hydroxyestradiol with the development of atherogenic diseases. [12] LPC (C16:0) caused apoptosis in human umbilical vein endothelial cells (HUVECs) and vascular smooth muscle cells, which can be related to atherogenesis [31]. In addition, estradiol induced eNOS activity in in vitro studies, showing rapid activity contributing to the great release of endothelial-derived nitric oxide [32]. Recent studies have also shown that ginsenosides have a positive effect on blood vessel health. Ginsenoside Rb3 alleviated the angiotensin II-induced decline of nitric oxide production and the phosphorylation of eNOS in HUVECs. Ginsenoside Rb3 increased endothelium-dependent relaxations, repressed endothelium-dependent contractions, and suppressed reactive oxygen species production from spontaneously hypertensive rats [29]. Ginsenoside Rb1 and ginseng water extract prevented HUVECs from oxidative damage-induced activation of eNOS pathways [33,34].

In addition to blood vessel health, ginseng intake has been shown to reduce the oxidative stress of the kidneys. The level of the renal oxidized form of glutathione, which is produced by oxidative stress and a known marker of oxidative stress in chronic renal failure [35], was decreased by ginseng intake in the present study. Although the activity of the related enzymes was not evaluated, this result partially supports the antioxidant effect of ginseng reported from aged rats [36] and astrocytes primary culture [37]. Unlike oxidized glutathione, the level of renal stearoylcarnitine was increased by ginseng intake. However, it is not known whether the reduction in stearoylcarnitine is positively related to kidney function since different trends have been reported depending on the specific disease. For example, rats with aristolochic acid nephropathy [38] had relatively decreased levels of stearoylcarnitine compared to those of normal rats, whereas the stearoylcarnitine level was increased in cardiovascular patients with end-stage renal disease [39].

## 5. Conclusions

Metabolomics and enzymatic analyses of the tissues and biofluids from ginseng extract-fed healthy male rats and the effects of ginseng intake on blood vessel tension using the aortas of the rats provide useful information to better understand the physiological effects of ginseng intake. In particular, ginseng intake decreased the levels of blood phospholipids, LPCs, LPE (18:3) and related enzymes, oxLDL, high blood pressure factors, and cytokines, while increasing blood vasodilation. Moreover, ginseng intake decreased the level of renal oxidized glutathione. Nevertheless, there are many limitations of this study, which should be considered in interpretation of these overall effects. Therefore, more in-depth studies are needed to better understand how ginseng intake regulates blood vessel health and renal oxidative stress, and the relationship between ginseng-affected steroid hormones and physiological changes in healthy animal models and humans. In addition, further investigation on the bioactive compounds of ginseng will be needed. Overall, our findings suggest that ginseng intake can improve blood vessel health via modulation of vasodilation, oxidation stress, and pro-inflammatory cytokines. Moreover, the decrease in renal oxidized glutathione indicated that ginseng intake is positively related to the reduction in oxidative stress-induced renal dysfunction.

## Figures and Tables

**Figure 1 nutrients-12-02238-f001:**
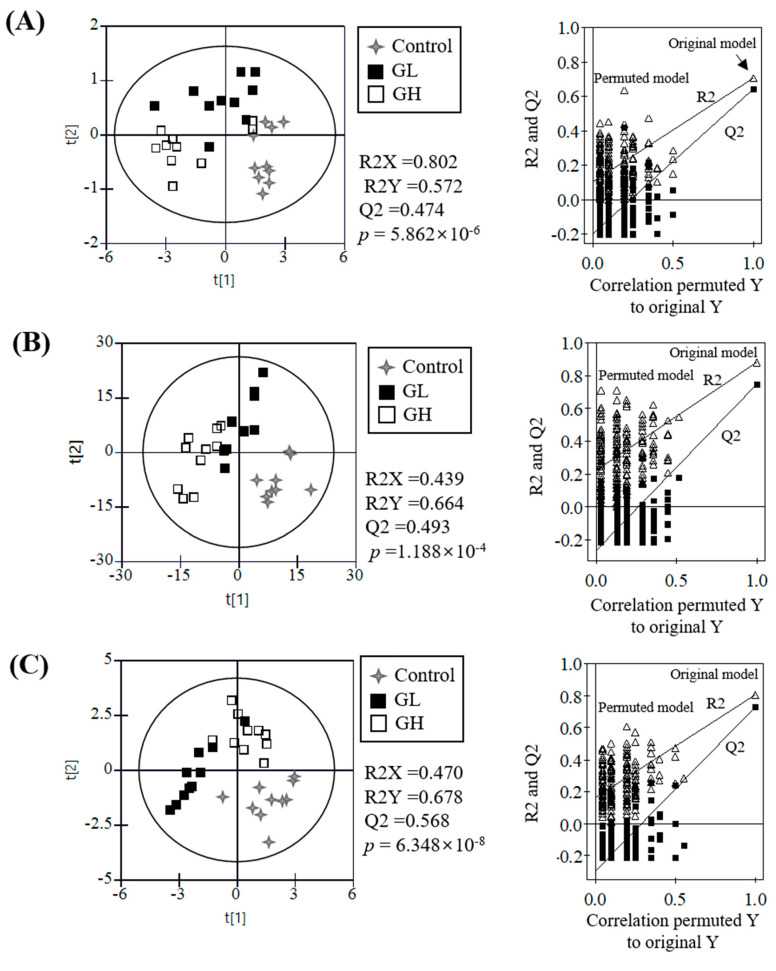
Partial least-squares discriminant analysis (PLS-DA) score plots of plasma metabolites (**A**), kidney metabolites (**B**), and plasma and urine steroid hormones (**C**). The qualification of the PLS-DA models was evaluated by R2X, R2Y, Q2, and *p*-values and validated by cross validation with a permutation test (*n* = 200). R2X and R2Y showed the fitting quality of the models and Q2 showed their prediction quality. Cross validation was evaluated by intercepts of R2X and R2Y and their final values.

**Figure 2 nutrients-12-02238-f002:**
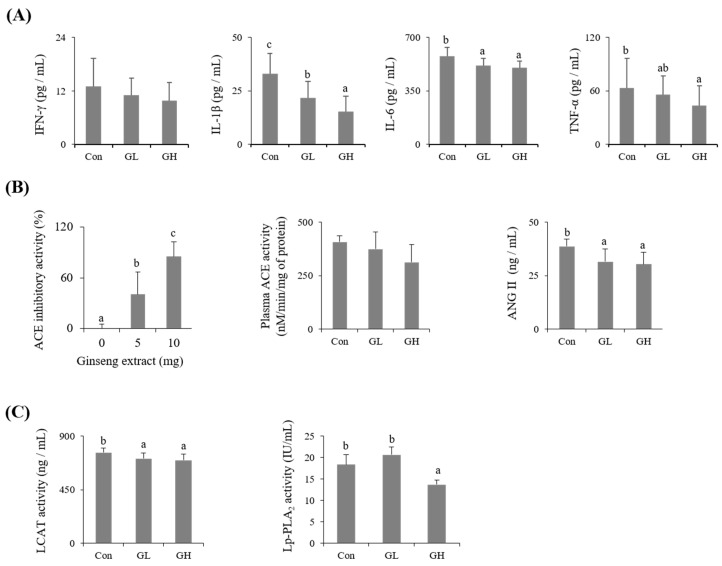
Effect of ginseng extract on production of plasma proinflammatory cytokines (**A**), blood pressure-related factors (**B**), and the plasma LPC-generating enzyme activities (**C**). Plasma Angiotensin (ANG) II, lipoprotein-associated phospholipase A_2_ (Lp-PLA_2_) and lecithin cholesterol acyltransferase (LCAT) were determined using enzyme-linked immunosorbent assay (ELISA) kits. Plasma inflammatory cytokines including IFN-γ, IL-1β, IL-6, and TNF-α were measured using a luminex screen assay kit. Angiotensin-converting enzyme (ACE) inhibitory activity of ginseng and plasma ACE activity were measured at 228 nm by a spectrophotometer. Different letters on the bars indicate d significant differences at *p* < 0.05.

**Figure 3 nutrients-12-02238-f003:**
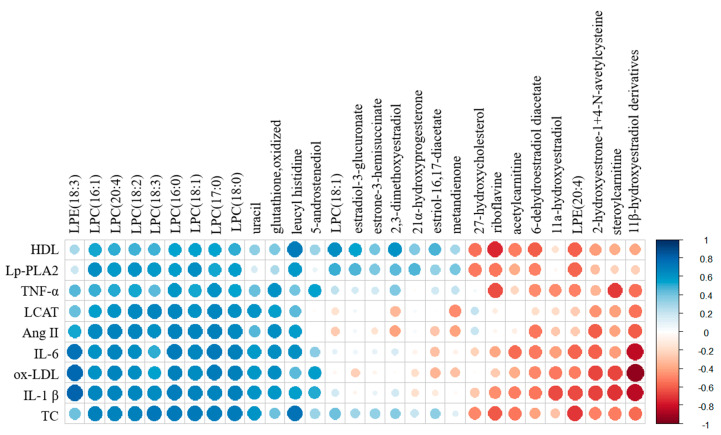
Analysis of correlations between identified metabolites and steroid hormones and blood health-related factors. The correlation matrix was analyzed and visualized with a heat map generated with the R corrplot package. Positive correlations are shown in blue, and negative correlations are shown in red.

**Figure 4 nutrients-12-02238-f004:**
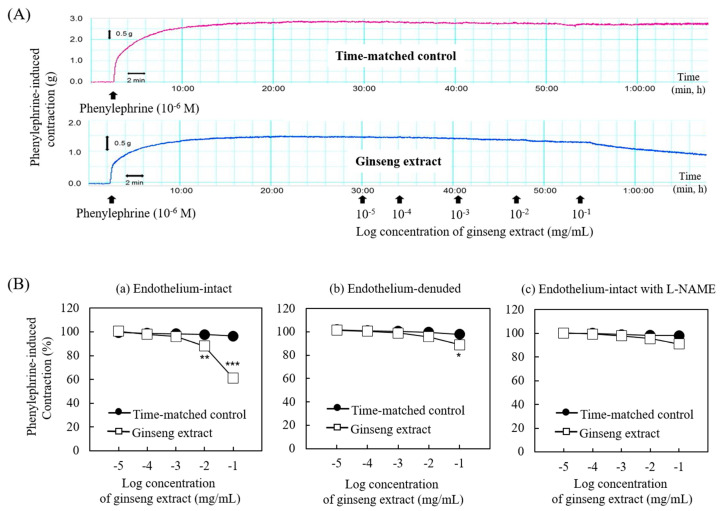
Traces showing the dose–response curves induced by cumulative addition of the ginseng extract in endothelium-intact aorta precontracted with phenylephrine (**A**) and the effect of ginseng extract on phenylephrine-induced contraction of the isolated endothelium-intact (a; *N* = 9) and -denuded rat aorta (b; *N* = 5 ) and isolated endothelium-intact rat aorta pretreated with N^W^-nitro arginine methyl ester (L-NAME) (c; *N* = 8) (**B**). Data are shown as mean ± SEM and expressed as percentage of maximal contraction induced by phenylephrine. N indicates the number of isolated rat aorta. * *P* < 0.05, ** *P* < 0.01, and *** *P* < 0.001 vs. time-matched control.

**Table 1 nutrients-12-02238-t001:** General characteristics of rats fed a normal diet with or without ginseng administration.

	Control	Ginseng	
		GL100 mg/kg/day	GH200 mg/kg/day
Body weight gain (g)	129.70 ± 15.74	123.90 ± 9.60	136.60±19.72
Food intake (g/day)	23.26 ± 0.66	23.98 ± 1.09	24.65 ± 1.15
Adipose tissue (g)	3.53 ± 0.40	3.43 ± 0.43	3.73 ± 0.59
Kidney (g)	1.11 ± 0.10	1.18 ± 0.09	1.14 ± 0.13
Liver (g)	10.53 ± 1.18	10.76 ± 0.98	11.06 ± 1.63
TG (mg/dL)	27.71 ± 1.70	24.00 ± 3.81	24.43 ± 4.47
TC (mg/dL)	122.17 ± 6.97 ^b^	116.8 ± 4.15 ^b^	101.17 ± 2.32 ^a^
HDL (mg/dL)	53.50 ± 3.02 ^b^	53.6 ± 2.30 ^b^	38.17 ± 3.43 ^a^
LDL (mg/dL)	64.37 ± 9.96 ^b^	58.4 ± 3.71 ^ab^	54.63 ± 3.90 ^a^
oxLDL (ng/dL)	17.48 ± 3.12 ^b^	10.35 ± 0.38 ^a^	10.56 ± 0.34 ^a^
oxLDL/HDL	0.33 ± 0.07 ^b^	0.19 ± 0.01 ^a^	0.28 ± 0.03 ^b^
oxLDL/LDL	0.28 ± 0.05 ^b^	0.18 ± 0.01 ^a^	0.19 ± 0.01 ^a^
oxLDL/TC	0.14 ± 0.02 ^b^	0.09 ± 0.00 ^a^	0.10 ± 0.01 ^a^
TC/HDL	2.29 ± 0.24	2.18 ± 0.10	2.67 ± 0.24

GL, 100 mg/kg of ginseng extract, GH, 200 mg/kg of ginseng extract; Values were expressed as mean ± SD (*n* = 10) and different letters in the same column indicated significant differences at *p* < 0.05. TG; triglyceride, TC; total cholesterol, HDL; high density lipoprotein cholesterol, LDL; low density lipoprotein cholesterol, oxLDL; oxidized low density lipoprotein.

**Table 2 nutrients-12-02238-t002:** Identified metabolites list obtained from UPLC-Q-TOF MS and identified sterol metabolites from multiple reaction monitoring (MRM) data of rats fed ginseng.

	Metabolite	*p*-Value ^a^	VIP ^b^	Fold Change (vs. Control)	
				GL	GH
Kidney	uracil	1.83 × 10^2^	0.94	−1.14	−1.19
	glutathione, oxidized	5.64 × 10^4^	1.40	−2.78	−2.87
	succinyladenosine	1.23 × 10^2^	0.95	−1.50	−1.00
	riboflavin	2.00 × 10^3^	1.21	−1.01	1.35
	leucyl histidine	2.51 × 10^5^	1.45	−1.02	−1.50
	stearoylcarnitine	2.70 × 10^2^	0.96	1.47	1.60
Liver	carnitine	3.30 × 10^2^	1.10	1.06	1.46
	LPE(C20:4)	2.40 × 10^3^	2.54	1.16	1.57
	LPC(C18:1)	2.24 × 10^2^	1.58	1.21	−1.33
Plasma	LPE(C18:3)	7.05 × 10^4^	0.69	−2.18	−2.66
	LPC(C16:1)	1.04 × 10^4^	0.55	−2.19	−3.97
	LPC(C20:4)	7.56 × 10^5^	1.00	−2.20	−4.50
	LPC(C18:2)	1.46 × 10^4^	2.48	−2.16	−4.62
	LPC(C18:3)	1.23 × 10^3^	0.17	−2.32	−3.39
	LPC(C16:0)	4.12 × 10^5^	3.28	−2.15	−3.93
	LPC(C18:1)	2.12 × 10^4^	1.88	−2.76	−6.58
	LPC(C17:0)	3.55 × 10^5^	0.49	−3.20	−7.34
	LPC(C18:0)	5.20 × 10^5^	2.41	−3.03	−5.90
	Estradiol-3-glucuronate	1.08 × 10^2^	0.96	1.10	−1.03
	estrone-3-hemisuccinate	9.92 × 10^3^	1.00	1.47	−1.30
	11α-hydroxyestradiol	2.62 × 10^23^	0.79	1.13	1.10
	21α-hydroxyprogesterone	3.87 × 10^3^	1.02	1.53	−1.08
Urine	27-hydroxycholesterol	1.78 × 10^2^	0.94	−1.25	1.19
	metandienone	3.76 × 10^2^	0.78	1.26	1.00
	2-hydroxyestrone-1+4-N-acetylcysteine	3.16 × 10^2^	0.90	1.98	2.13
	6-dehydroestradiol diacetate	2.71 × 10^2^	0.95	1.13	1.33
	estriol-16,17-diacetate	2.60 × 10^2^	0.82	2.14	−1.01
	2,3-dimethoxyestradiol	3.57 × 10^2^	0.85	1.48	−1.62
	5-androstenediol	4.17 × 10^2^	0.86	−1.42	−1.43
	11β-hydroxyestradiol derivatives	7.45 × 10^2^	1.86	6.75	6.67

^a^ Variable importance in the projection (VIP) values were determined by PLS-DA. ^b^
*p*-Value were analyzed by ANOVA with Duncan’s test.

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
