# Peer review of "Ginseng-Induced Changes to Blood Vessel Dilation and the Metabolome of Rats"

_nutrients, 2020, doi:10.3390/nu12082238_

Round 1

Reviewer 1 Report

Overall, the current manuscript lacks scientific significance by only demonstrating the correlational changes of few physiological traits with dietary supplements of ginseng extract in a rat model. Specifically, the authors did not delineate the effect of individual key ingredients from the whole ginseng extract and did not provide mechanistic studies on how each ingredient may lead to the observed changes in the cardiovascular system. Moreover, knowing the metabolic profiles and cardiovascular responses of the rat system is vastly different from the physiology of such in humans, the experimental designs along with the findings from this paper provided little clinical insight nor novelty.     

Author Response

We express our sincere appreciation for the reviewers’ many constructive suggestions for our manuscript.

Comment: Overall, the current manuscript lacks scientific significance by only demonstrating the correlational changes of few physiological traits with dietary supplements of ginseng extract in a rat model. Specifically, the authors did not delineate the effect of individual key ingredients from the whole ginseng extract and did not provide mechanistic studies on how each ingredient may lead to the observed changes in the cardiovascular system. Moreover, knowing the metabolic profiles and cardiovascular responses of the rat system is vastly different from the physiology of such in humans, the experimental designs along with the findings from this paper provided little clinical insight nor novelty.

Response: We agree that we did not investigate the effect of individual key ingredients from the ginseng extract on the blood vessel health. Moreover, we know that cardiovascular responses of the rat system is very different from the physiology of such humans. Although there are many limitations of this study, overall, our results provide a unique perspective on the influence of ginseng in not only improving cardiovascular risk factors but for maintaining health in the healthy population.

Reviewer 2 Report

This manuscript highlighted the effect of ginseng extract on the biofluids and tissue metabolites. Authors made a detailed study to reveal ginseng as a potential source for improving blood vessel health via modulation of vasodilation, oxidation stress, and pro-inflammatory cytokines.

Author Response

Response to Reviewer 2 comments

1. This manuscript highlighted the effect of ginseng extract on the biofluids and tissue metabolites. Authors made a detailed study to reveal ginseng as a potential source for improving blood vessel health via modulation of vasodilation, oxidation stress, and pro-inflammatory cytokines.

Response 1: Thank you for your review and comments on my manuscript.

Reviewer 3 Report

In this article, the authors addressed the role of ginseng consumption in reducing the risk of cardiovascular diseases. They found that ginseng intake decreases the levels of high blookd pressure factors, renal oxidized glutathione, and induced vasodilation. The authors showed strong and extensive evidence suggesting a role of ginseng intake in improving blood vessel health. There are still some minor points need to be addressed.

  1. All the labeling of Figure 2 and 4 are missing.
  2. All the names of metabolites are missing in figure 3. Although the supp fig.5 provide some information, it is very difficult to read.

Author Response

We express our sincere appreciation for the reviewers’ many constructive suggestions for our manuscript. Based on these comments, we have carefully revised the text to ensure consistency and clarity. 
